# Smart Window Based on Angular-Selective Absorption of Solar Radiation with Guest–Host Liquid Crystals

**Seong-Min Ji** †, **Seung-Won Oh** † and **Tae-Hoon Yoon** *

Department of Electronics Engineering, Pusan National University, Busan 46241, Korea; jisungmin@pusan.ac.kr (S.-M.J.); ohseungwon@pusan.ac.kr (S.-W.O.)
* Correspondence: thyoon@pusan.ac.kr
† These authors contributed equally to this work.

**Abstract:** In this study, we analyzed angular-selective absorption in a guest–host liquid crystal (GHLC) cell for its application in smart windows. For reducing the energy consumption, angular-selective absorption is desired because the light transmitted through windows during the daytime is predominantly incident obliquely from direct sunlight. Owing to the absorption anisotropy of guest dichroic dyes, a GHLC cell can absorb the obliquely incident light, while allowing people to see through windows in a normal view. Therefore, the cell can provide a comfortable environment for occupants, and reduce the energy required for cooling by blocking the solar heat incident from the oblique direction. The GHLC cell can be switched between the transparent and opaque states for a normal view. The rising (falling) time was 6.1 (80.5) ms when the applied voltage was 10 V.

**Keywords:** liquid crystal; switchable window; static window; selective absorption; energy saving

## 1. Introduction

Switchable windows allow users to control the sunlight entering a building and the solar heat throughput while allowing people to see through them; in other words, their view is not blocked even when the window is darkened [1–3]. Energy-saving, using these windows, has been widely studied to provide a comfortable environment for occupants, while saving energy for heating, cooling, and artificial lighting [4–9]. Over the past few decades, most studies on switchable windows have focused on chromic materials, such as electrochromic [10–13], photochromic [14–16], and thermochromic [17–20] materials, which change their color with external stimuli. Other switchable window technologies include liquid crystal (LC) devices [21–27] and suspended particle devices (SPDs) [28,29].

Unlike switchable windows, tinted windows absorb incident light from all directions, owing to their isotropic absorption properties. Therefore, regardless of their viewing angle, people always see a dark view through them. Thus, windows that can selectively regulate the directional light transmission with angle are practically significant for achieving optimal daylighting and indoor insolation. Windows with chromic materials have been proposed to realize the angular selectivity of incident light [30,31]. This requires complex multilayer structures because chromic devices absorb light from all directions with isotropic absorption properties.

In this study, we analyzed the angular-selective absorption of solar radiation in a tinted window made of guest–host liquid crystal (GHLC) cells. A GHLC cell can be a candidate for providing angular dependence (selective absorption) because the dichroic dye has absorption anisotropy. In particular, GHLC can absorb oblique incident light, while transmitting the light when it is incident normally. Furthermore, to provide a comfortable environment for people, a high transmittance difference between the normal and oblique views is required. Therefore, to determine the condition with a high transmittance difference between normal and oblique incident light, we calculated the path length and absorption coefficients of the dye, which affect the transmittance of a GHLC cell for oblique

incidence. Furthermore, we fabricated a GHLC cell and demonstrated that it selectively absorbs incident light by exhibiting angular dependence. Moreover, it can be switched from the transparent to opaque states in the normal view by applying the voltage as required.

## 2. Design Principles of the Device

Most studies concerned with the performance of light absorption windows have focused on normal incidence. However, when using these windows, we also need to consider the oblique incidence caused by varying solar elevation angles. This is because the intensity of light irradiated to windows during the daytime is mostly direct sunlight. Therefore, it is possible to save energy by controlling the transmission of direct sunlight entering the room, while also considering its seasonal variations. Figure 1a illustrates the seasonal elevation angle of the sun during the daytime. The solar elevation angle is highest at midday during summers. It is hence necessary to absorb sunlight as the incidence angle increases, to reduce the energy required for cooling. Conversely, at midday during winters, the solar elevation angle is relatively lower, and the temperature is significantly lower than it is in summer. Therefore, it is necessary to reduce the absorption of incident light to reduce the energy required for heating and lighting. In addition, because the solar elevation angle changes over time, the transmittance needs to be adjusted accordingly. A schematic of the window absorbing (or transmitting) the incident light in summer (or winter) is depicted in Figure 1b. As discussed, it is necessary to absorb (or transmit) the sunlight incident at a higher (or lower) elevation angle in summer (or winter) to save energy.

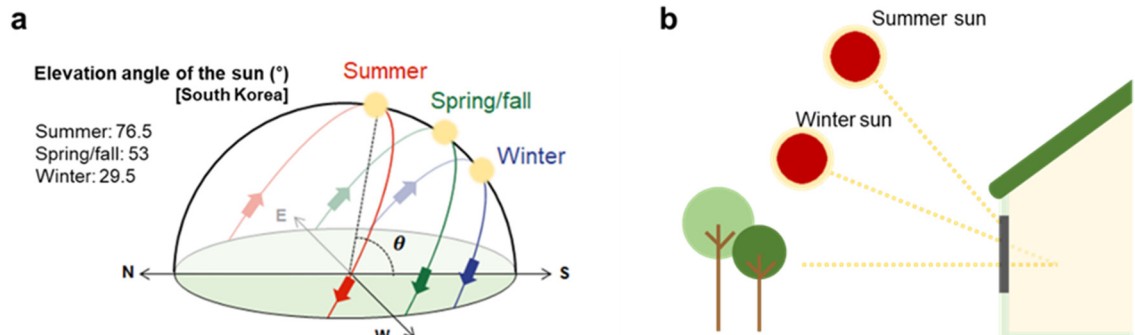

**Figure 1.** (**a**) Seasonal elevation angle of the sun in Korea. (**b**) Schematic of a tinted window in summer and winter.

GHLC devices, comprising a host LC and guest dichroic dye, have attracted considerable attention owing to their high transmittance in a transparent state and polarizer-free structure [32–41]. Due to their dichroism, dye molecules strongly (or weakly) absorb the light that is polarized parallel (or perpendicular) to their absorption axis (Figure 2a). Moreover, a GHLC device is convenient for switching because the dye molecules can be easily aligned by the rotation of LC molecules [41–46]. In the case of normal incidence, the transmittance of a homogeneously-aligned GHLC cell can be described as follows [47,48].

$$T_{\parallel} = T_0 e^{-\alpha_{\parallel} cd} \tag{1a}$$

$$T_{\perp} = T_0 e^{-\alpha_{\perp} cd} \tag{1b}$$

where, $T_{\parallel}$ [$\alpha_{\parallel}$] and $T_{\perp}$ [$\alpha_{\perp}$] are the transmittance (or absorption coefficients) of a GHLC cell for polarization parallel and perpendicular, respectively, to the absorption axis of the dye molecules. $T_0$ is the transmittance of a homogeneously-aligned LC cell without dye; $c$ and $d$ are the dye concentration and cell gap, respectively. Because the sunlight is unpolarized, the transmittance of a GHLC cell can be calculated as follows.

$$T = \frac{1}{2} T_0 \left( T_{\parallel} + T_{\perp} \right) \tag{2}$$

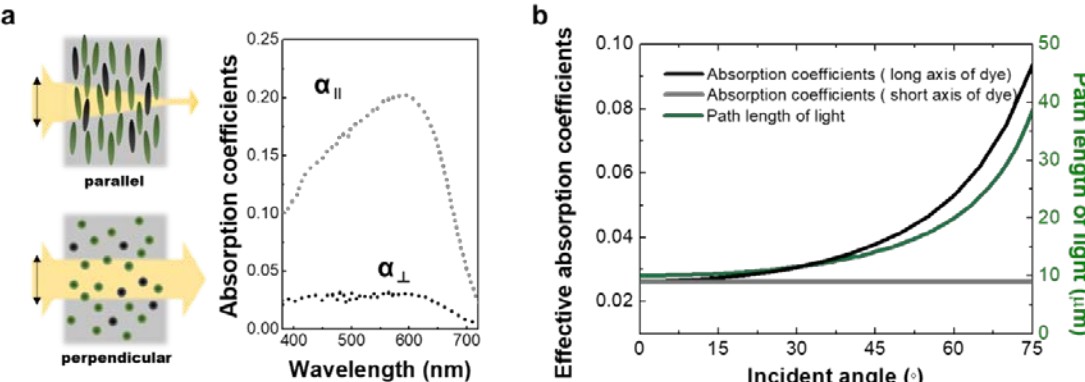

**Figure 2.** (**a**) Absorption coefficients of the black dichroic dye X12. (**b**) Calculated effective absorption coefficients and path length with increasing incident angle.

In the case of oblique incidence, $\alpha$ and $d$ in Equation (1) are dependent on the angle of incidence ($\theta_i$). Therefore, Equation (1) can be rewritten as:

$$T_{\parallel}(\theta_i) = T_0(\theta_i)e^{-\alpha_{\parallel}(\theta_i)cd(\theta_i)}, \tag{3a}$$

$$T_{\perp}(\theta_i) = T_0(\theta_i)e^{-\alpha_{\perp}cd(\theta_i)}, \tag{3b}$$

where $\alpha_{\parallel}(\theta_i)$ and $d(\theta_i)$ represent the absorption coefficient and path length of a GHLC cell with oblique incidence. $T_0(\theta_i)$ is the transmittance of a homogeneously-aligned LC cell without dye with oblique incidence. Furthermore, the effective absorption coefficient $\alpha_{\parallel}(\theta_i)$ can be calculated as follows [49].

$$\frac{1}{\alpha_{\parallel}^2(\theta_i)} = \frac{\sin^2\theta_i}{\alpha_{\parallel}^2} + \frac{\cos^2\theta_i}{\alpha_{\perp}^2} \tag{4}$$

where $\alpha_{\parallel}(\theta_i)$ increases with the increase in the incident angle. Here, we assume that the absorption axis of the dye molecules is oriented in the plane of incidence. The path length, $d(\theta_i)$, can be calculated as:

$$d(\theta i) = d/\cos\theta i \tag{5}$$

In other words, the path length $d(\theta_i)$ increases with increasing the incident angle. Therefore, the transmittance of a GHLC cell for oblique incidence can be calculated as follows.

$$T(\theta_i) = \frac{1}{2}T_0(\theta_i)\left(T_{\parallel}(\theta_i) + T_{\perp}(\theta_i)\right) \tag{6}$$

In Figure 2b, the calculated effective absorption coefficient and path length are plotted as functions of the angle of incidence. These two factors increased as the angle of incidence increased. Therefore, the transmittance of a GHLC cell could be decreased by increasing the angle of incidence.

To achieve a high transmittance difference between normal and oblique incidences, we found the optimum dye concentration and cell gap by following the optimization algorithm reported in [50], because they affect the transmittance of a GHLC cell. Depending on the applications, we could design a GHLC cell with a high contrast ratio by increasing the cell gap and dye concentration. Figure 3 presents the transmittance-difference contour map on the parameter space of the dye concentration and cell gap. For the numerical calculations shown in Figure 3, we used the physical parameters of negative LC (anisotropy $\Delta n = 0.1096$ and dielectric anisotropy $\Delta\epsilon = -3.9$ at 1 kHz) and 2 wt% of black dichroic dye X12 (see Figure 2a). We considered the elevation angle of the sunlight at midday during midsummer as the angle of oblique incidence because it is the hottest time of the season. The maximum value of the transmittance difference between the normal and oblique

incidences was 45%. Through the optimization process, we confirmed the factors that affect the transmittance of a GHLC cell and found the optimized conditions that are required to achieve a high transmittance difference between the normal and oblique incidences.

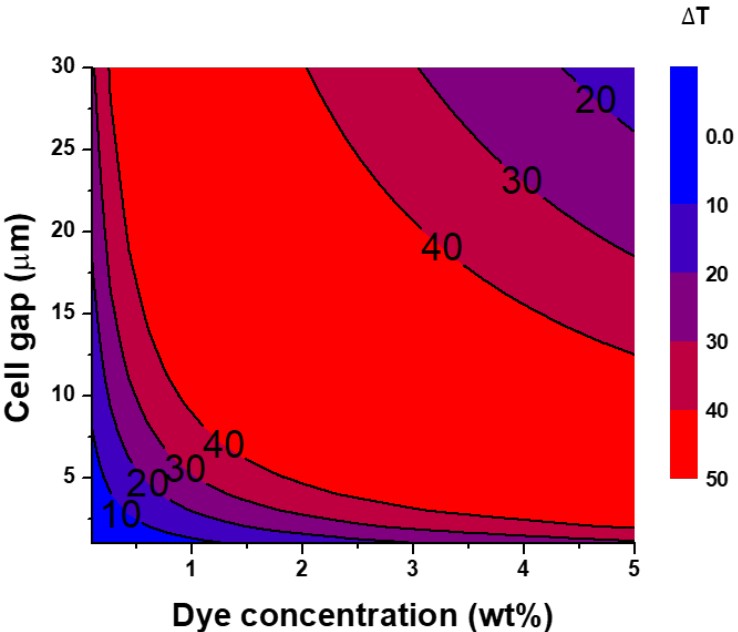

**Figure 3.** Calculated constant transmittance-difference contour map of a guest–host liquid crystal (GHLC) cell on the parameter space of the dye concentration and cell gap.

## 3. Device Fabrication and Experimental Results

We fabricated a GHLC cell based on the optimized conditions obtained from the transmittance-difference contour map depicted in Figure 3. To obtain a clear image for the normal view, the homeotropic alignment layer (AL64168, JSR Korea, Gongju, Korea) with the pretilt angle of 87.5° was spin-coated on the indium-tin-oxide-coated glass substrates. Spin-coating was performed for 10 s at 1100 rpm, and then 50 s at 4500 rpm. After the spin-coating process, the coated substrates were baked for 1 h at 250 °C. The thickness of the LC layer was maintained at 11 μm using silica-ball spacers. Negative LC ($\Delta n$ = 0.1096, $\Delta\epsilon$ = −3.9) was mixed with 2 wt% of black dye X12 (BASF, Ludwigshafen, Germany). The absorption coefficients $\alpha_\parallel$ and $\alpha_\perp$ of the mixture were 20.17 and 3.09 μm$^{-1}$ at 600 nm, respectively. The LC-dye mixture was obtained by stirring for 24 h at 40 °C. Finally, this LC mixture was injected into an empty cell via capillary action at room temperature. Figure S1 shows the polarized optical microscopy and conoscopic images of the fabricated GHLC cell. The dark POM image shows that the LC and dye molecules are aligned perpendicular to the substrates in the initial state. Crossed dark areas the conoscopic observation have also resulted from the homeotropic alignment of the LCs.

Numerical calculations of the transmittance of a GHLC cell with the change of the incident angle were performed using the commercial software tool "TechWiz LCD 1D" (Sanayi System Company, Ltd., Incheon, Korea), in which the calculation of the light transmission is based on the Berreman matrix method that automatically includes the refraction of the light. The transmittance of the fabricated GHLC cell was measured by changing the angle of incidence to confirm the angular dependence of light absorption. We used a xenon lamp, which emitted wavelengths ranging from 185 to 2000 nm (MC-961C, Otsuka Electronics Co., Ltd., Osaka, Japan) as a light source. We also measured the transmittance of the regular glass to compare it with that of the fabricated GHLC cell.

In Figure 4a, the calculated and measured transmittance of a GHLC cell and regular glass are plotted as functions of the angle of incidence. The transmittance of a GHLC cell

decreased with an increase in the incident angle. Conversely, the transmittance of a regular glass remained unchanged until the incident angle reached 45°.

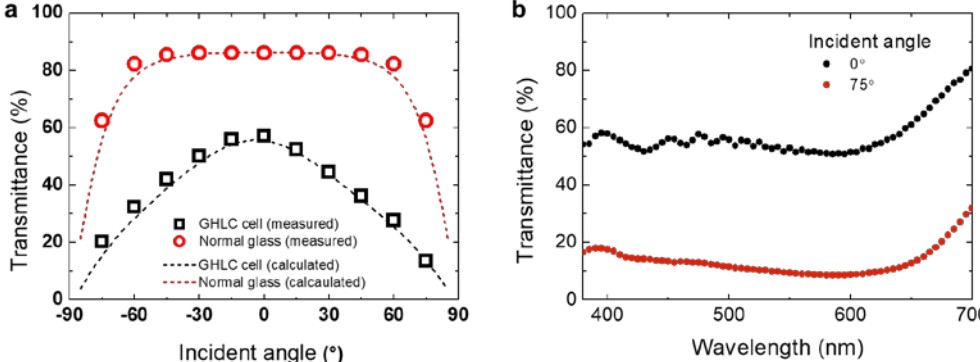

**Figure 4.** (**a**) Measured (circles and squares) and calculated (dash lines) transmittance of a GHLC cell and regular glass vs. the incident angle. (**b**) Measured transmission spectra of a GHLC cell under normal and oblique incidences.

Figure 4b shows the measured transmittance of the fabricated GHLC cell. The transmittance of the GHLC cell decreased from 57.0% at 0° to 13.4% at +75°. This proves that, as expected, a GHLC cell can selectively absorb incident light with angular dependence.

To show the angular dependence of light absorption, we provided the images of the fabricated GHLC cell and regular glass for comparison. Figure 5 shows the images of the fabricated GHLC cell and regular glass when the incident angle is changed from 0° to +75°. As the incident angle increased, the GHLC cell became darker while the regular glass remained white regardless of the incident angle.

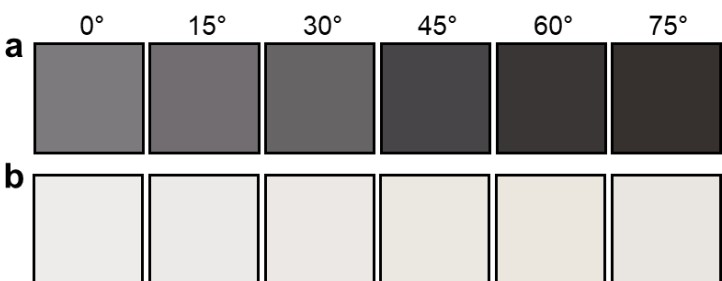

**Figure 5.** The images of (**a**) the fabricated GHLC cell and (**b**) regular glass at the incident angles of 0°, +15°, +30°, +45°, +60°, and +75°.

We analyzed the transmittance of the GHLC cell under normal and oblique incidence. Moreover, we analyzed its switchable aspect because the solar elevation angle changes throughout the day. We measured the transmittance of the fabricated GHLC cell by changing the angle of incidence as we applied a voltage to the cell (Figure 6). For switching to the on-state, a vertical voltage of 10 V was applied to the fabricated cell. The rising and falling times of the GHLC cell were 6.1 and 80.5 ms, respectively. Figure 6 shows that the transmittance of GHLC cell in the OFF state had a high angular dependence, whereas that in the ON state had a relatively low angular dependence. Thus, a GHLC cell could be utilized as a switchable window by applying a voltage as required.

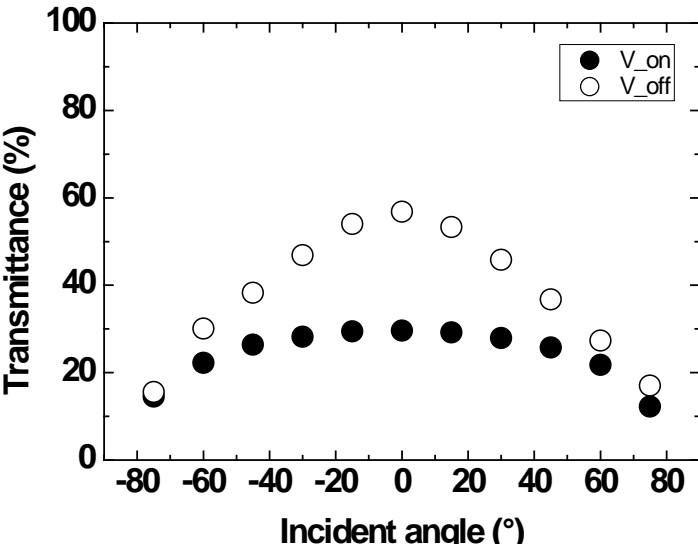

**Figure 6.** Measured transmittance of the fabricated GHLC cell vs. the incident angle in the ON and OFF states.

## 4. Conclusions

We analyzed a GHLC cell as a tinted window that can selectively absorb incident light by angular dependence. In particular, using dichroic dyes, which have absorption anisotropy, the light absorption window could absorb the obliquely incident light, while allowing people to see through it in the normal view. We confirmed that the transmittance of a GHLC cell under normal and oblique incidence was 57.0% and 13.4%, respectively. In addition, the GHLC cell could be switched by applying the voltage.

**Supplementary Materials:** The following are available online at https://www.mdpi.com/2073-4352/11/2/131/s1. Figure S1. Polarized optical microscopy image and conoscope observation of fabricated GHLC cell.

**Author Contributions:** S.-M.J., S.-W.O. and T.-H.Y. planned the study. S.-M.J. wrote the manuscript. S.-M.J. and S.-W.O. conceived and designed the guest–host liquid crystal cell. S.-M.J. and S.-W.O. performed the experiments. S.-M.J. and S.-W.O. analyzed the experimental and calculated data. T.-H.Y. supervised the analysis and co-wrote the manuscript. All authors discussed the results and commented on the manuscript at all stages. All authors have read and agreed to the published version of the manuscript.

**Funding:** This work was supported by the National Research Foundation of Korea (NRF) grant funded by the Korea government (MSIP) (No. 2020R1A2C1006464).

**Conflicts of Interest:** The authors declare no conflict of interest.

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
