# Peer review of "Smart Window Based on Angular-Selective Absorption of Solar Radiation with Guest–Host Liquid Crystals"

_crystals, doi:10.3390/cryst11020131_

Round 1

Reviewer 1 Report

Review Report

In this manuscript, the authors report a smart window based on the polarized absorption of the dichroic dye. The guest-host proposed here is not novel, but the smart window with blocking heat might be useful. Unfortunately, the characterization of the device is not complete to warrant publication of the manuscript in its current form, and major revisions are required before it can be considered for further publication in Crystals.

  1. The novelty is questionable since both of the GHLC has been well studied.
  2. The measured contrast ratio between the normal incidence and 75° incidence is around 60%/20%=3, could it be further improved with some potential solutions?
  3. What is the response time between the on and off state? What is the voltage?
  4. Could the authors please add some POM pictures to provide the readers with a more comprehensive understanding of the structure?
  5. In Figure 5, the contrast of the photos in Figs 5(e) and (f) seems to be different from other pictures. Are these two original pictures or a picture that has been slightly edited? If they are the original pictures, could the authors please add some discussions or explanations about the obvious yellowish compared with the other four; otherwise, could the authors please use to original pictures to make a more fair comparison?

Reviewer 2 Report

The manuscript reports on using the linear dichroism of a dye-doped liquid crystal to design a switchable window that naturally responds differently to different seasons. Although the mechanism is well known and widely investigated, the design itself is interesting. This work is recommended for publication after addressing the following comments.

(1)   It is difficult to confirm if Eq. 4 is a valid treatment. Please specify the chapter and section of Ref. [43]. 

(2)   The Snell’s law of refraction has a strong effect on the interaction angle between optical polarization and dye orientation and is not considered in Eq. 5 and simulations (Figs. 2,3,4, and 6). The simulations should therefore be corrected.   

(3)   [continued] Taking the typical average index of a liquid crystal to be ~1.5, the maximum incident angle in the LC layer is only ~40 degrees (rather than 90 degrees). Considering optical refraction, is the contrast between large and small incident angles still significant as claimed by the authors?

(4)   The setting for the simulation shown in Fig. 3 should be given.  

(5)   The dye material and its properties (e.g., two absorption coefficients) should be given.   

(6)   Additionally, it is worth mentioning some recent advances of LC-based smart windows in the introduction, especially those related to dichroic dyes. Besides authors’ other publications cited as Refs. [32–37] and [40], following are other recommended papers:

(i) "Versatile Energy-Saving Smart Glass Based on Tristable Cholesteric Liquid Crystals" ACS Appl. Energy Mater. 3, 7601–7609 (2020)

(ii) "A Single‐Step Dual Stabilization of Smart Window by the Formation of Liquid Crystal Physical Gels and the Construction of Liquid Crystal Chambers" Adv. Opt. Mater. 30, 1906780 (2020)

(iii) "Smart Window with Active-Passive Hybrid Control" Materials 13, 4137 (2020)

(iv) "Infrared Regulating Smart Window Based on Organic Materials" Adv. Energy Mater. 7, 1602209 (2017)  

Reviewer 3 Report

The abstract is unsatisfactory. Only general and qualitative descriptions are provided without any numbers, which is neither informative nor convincing to the target readers. How big is the applied voltage? How is the performance? How fast is the switching? All these key performances are unclear from the abstract. The authors must add quantitative results for the key findings, including comparative results with states of the arts.

Some of the references are outdated. The introduction and literature survey sections should be better developed by reviewing more up-to-date publications on liquid crystals based technologies, e.g. A. Yontem, J. Li, and D. Chu, “Imaging through a projection screen using bi-stable switchable diffusive photon sieves,” Opt. Express, vol. 26, pp. 10162–10170, April 2018. doi: 10.1364/OE.26.010162

The structure of the manuscript needs to be better organized. Some sections are extremely short, e.g. section 3 Device fabrication. It is advisable to merge with the experimental results section.

Even though the fabrication is following a standard procedure - it is advisable that the authors describe with sufficient details, with sufficient information to be reproducible by another investigator. For instance, spin-coating speed, how did you prepare the mixture in what conditions, angle of the homeotropic alignment layer, capillary action at what temperature, etc. Too many details are missing here. Using a flow chart would be helpful.

In addition, the authors must mention the Table or figures before showing them, for example, Figure 3. The contour map in Figure 3 is suggested to use colorful pictures to agree with other images you presented.

The authors claimed to have performed optimizations, e.g. "optimum dye concentration", "optimization process". What optimization algorithm or strategy do you use? Optimization is a competition of different factors (that are usually in contradiction with each other) and finding the optimal balance. To claim as optimization, the authors must add sufficient quantitative details to justify. Figure 3 may not be sufficient.

Moreover, optimization is normally combined with experimental work, as simulated optimal design is generally not the experimentally optimum. The fabrication is only based on the "optimized" conditions by calculations. The authors should continue with experimental optimization work and complete the study, more iterations of designs are required.

Round 2

Reviewer 1 Report

The revised manuscript provides more information about the optical properties and parameters. All questions have been answered well by the authors. Although I personally remain comments on my previous point 1, the revised vision of the work has the potential for publication.

Reviewer 3 Report

After the previous round of review and revision, the major problems have been substantially addressed.

The newly added details have some minor issues to fix.

"The rising [falling] time was 6.1 [80.5] ms when the applied voltage was 10 V." - It is not a standard way to use []. () is preferred. 

"physical parameters of negative LC (anisotropy ∆n= 0.1096 and dielectric anisotropy ∆ϵ = -3.9)" - Please justify the parameters are at what frequency.

"The absorption coefficients α∥ and α⊥ of the mixture were 20.17 μm−1 and 3.09 μm−1 at 600 nm, respectively." - Formatting of the unit is problematic. 

The newly added Figure 5 does not look like an experimental photo. 
